# More popular because you're older? Relative age effect on popularity among adolescents in class

**Danelien A. E. van Aalst**[1]*, **Frank van Tubergen**[2]

1 Department of Sociology, University of Groningen, Groningen, Netherlands, 2 Department of Sociology, Utrecht University, Utrecht, Netherlands

* d.a.e.van.aalst@rug.nl

## Abstract

Previous studies have found a significant effect of pupil's month of birth on their school- and sports performances. The current study investigates whether this so-called relative age effect also exists in a rather unexplored domain, namely popularity among adolescents in school classes. Whereas prior studies examined relative age related to the cut-off date at primary school entry, we also study possible relative age effects regarding the age composition within pupils' current school class. Data are from nationally-representative surveys among 14–15 year-old pupils from the Netherlands, Sweden and England. Results indicate a statistically significant positive relation of both types of relative age with popularity status in classes. The relation of past relative age is particularly strong in England, which has a system of social promotion, whereas current relative age is strong in the Netherlands, with its system of grade retention. These findings underscore the importance of education policy.

## Introduction

Most Western countries make use of systems with a minimum age and cut-off dates for school entry and youth sport teams [1]. Consequently, there is an age difference up to 12 months between pupils within one school class or team, leading to accompanying cognitive, social and physical differences between children born just prior to the cut-off date and those born right after [1, 2]. This phenomenon is called relative age and its consequences are known as the Relative Age Effect (RAE) [3].

Previous research on RAE has found negative consequences for relatively younger pupils compared to their older peers in education and sports. Relatively younger pupils were found to have lower test scores in math, science and in reading- and writing skills and higher chances of being evaluated and diagnosed with learning disabilities than their older peers [1, 4–6]. Relatively younger pupils are also overrepresented in lower school tracks [4, 5], less often enrolled in college [7], and more likely to repeat a grade [8, 9]. A second strand of research focused on the presence of RAE in sports and showed substantial disadvantages for relatively younger sports players; they were overrepresented in sports teams at the lowest level in many sports

**Data Availability Statement:** The full version of this dataset is available via the GESIS Data Archive at https://doi.org/10.4232/cils4eu.5353.3.3.0 The codebook, questionnaires and reports can be found here.

https://www.cils4.eu/index.php?option=com_content&view=article&id=23&Itemid=18.

**Funding:** The author(s) received no specific funding for this work.

**Competing interests:** The authors have declared that no competing interests exist.

such as ice hockey, soccer and football. The effect was found at all ages, from early childhood to late adolescence and in professional sports teams [3, 10, 11].

More recently, RAE has also been investigated in other domains. One study on leadership experience [6] showed that relatively younger pupils gain less leadership experience during high school, which can negatively affect labour market outcomes such as wages and the likelihood of becoming a CEO. With regard to psychological wellbeing, another study [7] found that entering primary school at a relative young age is associated with lower self-esteem later on. Furthermore, prior work found that those who were born right before the school cutoff day have higher mortality rates by suicide, compared to their peers who were born right after the cutoff date [12].

The current study contributes to the field by testing RAE in a rather unexplored domain, namely popularity of adolescents in school classes. As adolescents are highly sensitive to status at this age, it becomes very important to achieve or increase status [13]. Status can be derived from personal preference of an individual student of its peers, indicated by liking a peer or seeing him as a friend, also called social preference. On the other hand, there is a more reputation based status, that refers to which students are perceived by their peers as being most popular, influential and powerful [14]. Popularity and social preference can thus be seen as two dimensions of social status. Popularity, however, is more based on how a student is perceived by its peers based on a combination of peer-valued characteristics, for example athletic skills, physical appearance, attractiveness, combined with social and cognitive skills, and sometimes antisocial or deviant behavior such as smoking and aggression [13]. Popularity is defined as the relative position of an individual in the peer hierarchy and identified as an important value for all human beings [15, 16]. The peer hierarchy comes into existence by the group assigning status to its members [17] and can be viewed as a continuum where those high in the hierarchy are very popular, powerful and have influence, while those lower in the hierarchy are more likely to be victimized and isolated [15].

Especially in the beginning of secondary education, an age difference of 11 months between the oldest and youngest student in class could mean a significant difference in physiological development, for instance size, strength, and sexual maturity, and cognitive development, for instance neurocognitive functions that become more efficient as children grow older and social skills [1, 3]. Students who are relatively older thus have several advantages over their younger classmates in achieving and maintaining popularity status. It is reasonable to expect that these adolescents are more likely to be perceived as powerful, influential and, more self-confident. The latter being in line with the Pygmalion effect, where higher expectations of the teacher and other adults result in higher performance and more self-confidence [3], which would benefit their popularity ratings.

A second contribution to the literature is that we disentangle two types of RAE. Like previous research on RAE, the current study also uses the month of birth in relation to the prevailing cut-off date to test the accumulated effect of relative age. This is what we call *past relative age*, and this starts as early as primary school (typically at age 4–6) [5]. However, next to investigating past relative age, this study introduces a new theoretical concept, namely *current relative age*. This refers to the relative age of each pupil within the age composition of their current class, whereby each pupil is assigned an exact age position in relation to classmates. Since past relative age is also taken into account, the second contribution of this study is to distinguish the two effects, thereby providing insight in the duration of past relative age.

A third contribution is that we study the impact of education policies. Previous work on RAE has primarily focused on single country studies, which means that little is known about the role nationwide policies may have. The current study relies on cross-national data to

investigate how institutional differences between systems of grade retention and social promotion may impact the influence of past and current relative age on popularity.

The research question that will be answered in this study reads: To what extent are past and current relative age related to popularity of adolescents in school classes? To answer this question, we use nationally-representative data on more than 13,000 adolescents aged 14–15 in the Netherlands, Sweden and England.

## Hypotheses

Previous studies have argued that pupils strive to become popular in school and in order to achieve that, they draw upon their resources [17, 18]. These resources are formed by performance in sports, physical strength and appearance, cognitive skills and self-esteem [19–21].

Applying the theoretical framework on RAE on popularity of pupils, allows us to derive several hypotheses. First, by looking at *past* relative age, thus at entering primary school, we assume that relatively younger pupils are physically less strong and perceived as less attractive and possess fewer cognitive skills than older classmates. Entering primary school at the age of four to six (depending on the country), the difference between pupils due to age-based grouping, could go up to around 20% age difference between the youngest and oldest pupil [5]. This introduces systematic differences in maturity, cognitive skills [6] and the extent to which pupils are receptive for learning [22] with relatively younger pupils being disadvantaged [6]. These disadvantages probably have negatively affected their level of self-esteem and the self-perception of their skills, resulting in fewer resources for relatively younger pupils to draw upon in social interaction [20], which are needed in order to become popular in class and attain a high position in the peer hierarchy [17]. The first hypothesis therefore reads: The younger a pupil is with regard to the cut-off date at primary school entry, the less popular (s)he is in high school (H1).

Furthermore, this study focuses on a second type of relative age, namely the *current* age position of the pupil within the school class. We study high schools, and the transition into this social environment creates new opportunities to compete for status. Especially during adolescence, becoming popular among peers in class becomes an important goal [16]. A typical school class in high school consists of 20–30 children who attend classes together during the school year. The age composition can vary across classes within the same school, which implies that the same child born in, say, December might be the youngest in one school class, but not so in another school class. We assume that the same mechanisms work out, namely that those who are relatively younger with regard to the age composition of the school class have fewer resources to draw upon in becoming popular. This results in the second hypothesis: The younger a pupil is with regard to age composition in the current school class, the less popular (s)he is in class (H2).

We argue that the impact of past and current relative age depends on institutional conditions. Some countries have a system of grade retention, whereas other countries have a system of social promotion. Class repetition means that a pupil begins a new academic year in the same grade as the previous year [23]. Grade repetition may occur when pupils have not passed certain requirements or examinations that test the minimum level of educational attainment that should be achieved in the specific grade. If the pupil does not fulfil or pass these, the teacher will recommend grade retention [24], with the intention to give the pupil some time to mature and offer a chance to catch up on the academic level [25, 26]. In a system of social promotion, students are instead promoted to the next grade after the current school year, regardless of their achievements.

The institutional differences between systems of grade retention and social promotion may impact the role of past and current relative age on popularity. It has been found that in countries that have a policy of grade retention, relatively younger pupils are more likely to repeat a class [9, 23, 25]. This means that an initial disadvantage of past relative age may switch to a current relative age advantage: pupils who were relatively young in their former class, may suddenly be among the relatively oldest pupils in their new class, because of grade repetition. Such a "flipping" mechanism does not occur in countries with a policy of social promotion. Furthermore, in countries with a system of grade retention and skipping, the age range of students in the classroom is larger than in countries which have a system of social promotion. We assume that larger age differences in classrooms make age a more salient social category, and thereby become a more important element of popularity nominations.

The current study examines one country that has a system of grade retention (the Netherlands) and two countries that have a policy of social promotion (England and Sweden) [23]. Such official policies need not always be implemented in practice, of course. Indeed, in the nationally-representative data which we use for this study (see below), we find that among the 14–15 years old, around 5% have repeated a grade in Sweden. Thus, although Sweden officially has a system of social promotion, in some cases pupils can repeat a grade. But it happens much less frequently than in the Netherlands, where, drawing on our data, around 24% of the pupils have repeated a grade at age 14–15. In England, grade repetition is not happening.

We assume that past relative age more strongly relates to current relative age in countries that have systems of social promotion rather than grade retention. We therefore hypothesize that, compared to the Netherlands, among pupils living in England and Sweden the impact of past relative age on their popularity is stronger (H3a), whereas the impact of impact of current relative age on their popularity is weaker (H3b). To keep the comparison between the two systems as sound as we can, we will remove the respondents in Sweden who reported that they had repeated a grade.

When a pupil repeats a grade, it will lead to a higher ranking in the current relative age position, and thereby increase his or her popularity. Grade repetition, however, is also related to other mechanisms, which may impact popularity. Studies reveal that grade retention increases self-esteem in the short run, but that the effect diminishes over time and may undermine self-esteem in the long run [24]. Other research related grade repetition to behavioural problems and aggression [23, 24, 26]. Because these mechanisms may work out in opposite direction, their combined impact on popularity is difficult to predict. We therefore explore empirically the impact of grade retention on popularity.

## Methods

### Materials

The current study uses data from the first wave of the Children of Immigrants Longitudinal Survey in Four European Countries (CILS4EU) [27]. This nationally-representative survey was conducted at high schools in four European countries; England, Germany, The Netherlands and Sweden. The data were collected between October 2010 and April 2011 among adolescents in the age of 14–15 years, whereby children of immigrants were oversampled [27]. Germany was excluded in the current study, because it had not a clear cut-off date among the cohort of pupils in the data.

The data were collected in two steps. First, using a national register of schools, different strata were distinguished by their share of pupils with an immigrant background. From this, a random sample of schools were drawn in each country, oversampling schools with a higher share of immigrant youth. If a school did not participate, it was replaced with another school

with the same characteristics. This resulted in schools' response rates of 65.6% for England, 91.7% for the Netherlands and 76.8% for Sweden [27].

In the second step, (mostly) two school classes in the third year of high school (i.e., when pupils are around 14–15 years old) were randomly selected within each participating school. The classes that were selected were self-contained classrooms, i.e., the same group of 14–15 years olds stayed with the same group of peers for the school day. They follow all (or almost all) lessons with the same group. All pupils within these classes were invited to fill in a paper-pencil questionnaire during school hours. The student response rates were 81% in England, 86% in Sweden and 91% in the Netherlands. An ethical review of the CILS4EU project had not been carried out prior to data collection (2010–2011), but because of new EU regulations (GDPR) since 2016, the CILS4EU project was submitted to the Ethics Committee of the Faculty of Social and Behavioural Sciences Utrecht University. This Ethics Committee approved using the CILS4EU data for scientific purposes (FETC20-610). Participation was voluntary, and at the time of data collection, students were around 14–15 years of age. Because of their younger age, both the students and their parents were informed about the aims and design of the research. In each of the three countries for which we use data in this study (i.e., Sweden, Netherlands, England) both the students and their parents had the option to opt-out from the study. Data were therefore only collected among students who agreed to participate in the study, and whose parents agreed as well.

From the initial sample (13,703), 3% were excluded from the analysis (i.e., 452 pupils): 75 pupils were removed as they had missing values on birth month, 71 pupils missed data on year of birth which we used to calculate current relative age, 9 pupils had a missing value on sex, and 297 Swedish pupils were removed who repeated a class. The final sample size was 13,251. Table 1 provides the final sample sizes for each country.

The CILS4EU study aimed to compare majority-minority group comparisons with regard to intergenerational integration. Children, and their teachers and parents, were interviewed and then followed over the next two years. Information was collected on structural, cultural and social integration and identification with the home country. The main questionnaire contained information on date of birth and gender, while the sociometric part provided information on popularity nominations. A more extensive description of the design, sample selection and data collection of the CILS4EU study can be found elsewhere ([27–30]; see website https://www.cils4.eu/ for all data and reports).

The CILS4EU data are particularly suitable for this study for several reasons. First, the data contain sociometric data on relations within classrooms (average of ~22 pupils per class), from which a measure of popularity status can be derived for each pupil in class [31]. A second advantage is on the information on birth date for students. Most previous research used only month or season of birth as an indicator of relative age [1, 22]. The CILS4EU data contain information on both month and year of birth for pupils from all for countries, creating the possibility to derive two variables and distinguish between past and current relative age.

**Table 1. Final sample size.**

|             | Schools (N) | Classes (N) | Students (N) |
|-------------|-------------|-------------|--------------|
| England     | 107         | 208         | 4.223        |
| Netherlands | 100         | 222         | 4.310        |
| Sweden      | 129         | 251         | 4.727        |

## Measurement

**Popularity.** The dependent variable in the model is popularity status of the pupil in the peer hierarchy. Pupils were asked to nominate up to five popular classmates, but could, however, also decide to nominate no one or less than five pupils. Pupils in all countries were asked the same question in their own language, namely "Who are the most popular students in this class? Here you may write down no more than five numbers". The question asked to students was similar in all three countries, and nominations were based on provided class lists where each student was assigned a random number. Students were asked to write down the numbers corresponding to the student in answering the questions.

We are aware that at secondary schools, students also have friends outside their classroom, and outside their school, for instance from their sports club or neighborhood. However, we focus here on the popularity hierarchy and age distribution *within* their self-contained classrooms. We therefore believe this to be an accurate, and widely used, measure for popularity of adolescents [13].

A variable was created reflecting how often each pupil was nominated by his classmates. In order to take the differences in class size into account, the absolute numbers were translated into percentages, yielding scores from 0 to 100, where 0 means that the pupil was not nominated by any of his classmates, and 100 means that all classmates nominated the pupil for being popular in class. We constructed the variable as $(\sum_i B_{ji}/(N-1)) * 100$, where $i$ is the pupil, $B_{ji}$ indicates whether pupil $i$ was nominated by classmate $j$ for being popular and $N$ is the total number of pupils in class. An important issue is whether this sociometric measure has the same meaning and correlates in the three countries. Previous studies, using similar sociometric measures, found an association between popularity and bullying in the Netherlands [32–34] and Sweden [35], and also in other countries like Canada [36] and the United States [37, 38]. We could not find studies on England that relied on sociometric measures, but given the cross-national evidence for other countries, we assume that the sociometric measure of popularity has the same meaning in England.

**Past relative age.** The survey was conducted in 2010 when most pupils were around 14–15 years old. Depending on the country, the children entered primary school at different ages and countries use different cut-off dates. In England, the cut-off date at primary school-entry at the time these pupils entered primary school was September 1st [39]; in the Netherlands it was October 1st [40], and in Sweden January 1st [41].

The *past* relative age was therefore constructed by considering the month of birth for each pupil compared to the prevailing cut-off date of the country of residence. Pupils born in the month of the cut-off date were assigned a score of 11 and for every subsequent month after the cut-off date 1 was subtracted, resulting in a range from 11 to 0. A higher score indicates a relatively older pupil at school entry. The number of cases and age range per birth month after cut-off date is presented in S1 Table.

**Current relative age.** The current relative age captures the age of each pupil relative to the age composition in the school class at the time of the survey, i.e. when they were in high school and aged 14–15. While for the Netherlands and Sweden, the day, month and year were known, for comparability purposes we only used month and year of birth for all pupils from all three countries. Pupils were classified based on month and year of birth and pupils from the same class that were born in the same month and year were assigned the same value, by calculating the mean over the unique values that were initially assigned to these pupils. Specifically, we capture this idea in the following way: $C_i = \frac{C_1 + C_2 + \cdots + C_n}{n}$ where $C_i$ is the final age position of the pupil in class, $C_n$ is the unique value each pupil received and $n$ is the number of pupils with the same month and year of birth in class.

As not all classes contain the same number of pupils, class size was taken into account. Formally the construction of the final variable can be defined as: $(C_i/(N-1))$, where $C_i$ is the age position of the pupil $i$ in class and $N$ is the total number of pupils in a school class. This results in a range between 0 and 1, with a higher score indicating that the pupil is relatively older compared to classmates.

**Repetition.**   To examine the impact of grade repetition, we study pupils in the Netherlands only, because in both England and Sweden a system of social promotion exists. The data we use indeed show no repetition among pupils in England, but, as mentioned, we still found some Swedish pupils who repeated a grade (5%). The data for Dutch students contained information indicating whether or not a pupil has repeated a class in primary school, secondary school or both. The survey does not contain information on skipping classes, but we suspect that figures would resemble those of grade repetition, i.e., skipping classes being (virtually) absent in England, uncommon in Sweden, and more often occurring in the Netherlands. Note that, in line with our assumptions, it appears that the age range in years in classrooms are larger in the Netherlands ($SD_{age}$ = .597) than in England ($SD_{age}$ = .377) and Sweden ($SD_{age}$ = .344).

The data we use are hierarchical, i.e., students (level 1) are nested in classrooms (level 2), which are nested in schools (level 3). At each level, we include control variables. At the individual level, we control for gender. At the classroom level, we take into account the size of the class (i.e., the number of pupils), because pupils could nominate no more than five peers as popular, which means that chances to be nominated as popular decrease with class size. In addition, we include the mean popularity per classroom, i.e., the total number of popularity nominations in the class divided by the number of students. In this way, we take into account that the thresholds for popularity nominations may vary across classes. Finally, at the school level, we include the immigrant stratum, i.e., the percentage of the school with an immigrant background.

Table 2 provides summary statistics for all variables that were included in the models.

## Results

Fig 1 presents graphs that show the average popularity scores relative to birth months in three countries. The vertical line indicates the cut-off date, which differs across countries. Left of the dashed line are children born in birth month associated with being younger at primary school entry, whereas to the right side, children are older. For example, the cut-off data in England is 1st September, which means that children born in the period September-February are to the right side of the vertical line, and those born March-August to the left side. In England, therefore, children born in September are 11 months older than their peers in the school class who were born in August. As can be seen, there is quite some heterogeneity in the popularity scores within the groups before and after the cut-off date. This means that a regression discontinuity design [9, 12] which distinguishes treatment (i.e., those born after the cut-off date) and control (i.e., those born before the cut-off date), would be less appropriate in this case, because it does not capture this heterogeneity in the control and treatment groups. In particular in England, but also in Sweden and the Netherlands, the popularity ranking appears to drop more or less linearly with each month distance to the "favourable" birth month. Such a tendency can be captured by the more-refined measure that we include of past relative age.

Given the hierarchical nature of the data (schools, classrooms, pupils), we estimated three-level random intercept models. We used hierarchical linear regression, because of various desirable properties of linear modelling [42]. All analyses were done with Stata 16, using the *xtmixed* command. Because the dependent variable is a share or proportion, having a skewed

**Table 2. Descriptive statistics of dependent and independent variables.**

| Variable | Mean (proportion) | SD | Range |
|---|---|---|---|
| **Popularity** | 9.886 | 15.392 | 0–100 |
| **Class size** | 21.742 | 4.473 | 4–40 |
| **Class popularity (mean)** | 1.917 | 0.842 | 0–4.938 |
| **Past relative age** | 5.580 | 3.443 | 0–11 |
| **Current relative age** | 0.496 | 0.300 | 0–1 |
| **Country** | | | |
| Netherlands | 0.325 | | 0/1 |
| Sweden | 0.356 | | 0/1 |
| England | 0.318 | | 0/1 |
| **Gender** | | | |
| Boy | 0.496 | | 0/1 |
| Girl | 0.504 | | 0/1 |
| **Grade repetition** | | | |
| None | 0.850 | | 0/1 |
| Primary | 0.102 | | 0/1 |
| Secondary | 0.042 | | 0/1 |
| Both | 0.006 | | 0/1 |
| **% immigrants in school** | | | |
| 0–10 | 0.163 | | 0/1 |
| 10–30 | 0.327 | | 0/1 |
| 30–60 | 0.270 | | 0/1 |
| 60–100 | 0.212 | | 0/1 |
| Independent school | 0.027 | | 0/1 |

distribution, we also estimated fractional regression models [43] (S2 Table). The results from these additional analyses lead to the same substantive conclusions.

The results in Table 3 show that past relative age had a statistically significant positive association with popularity (Model 1). To illustrate, the coefficient for the Netherlands (reference

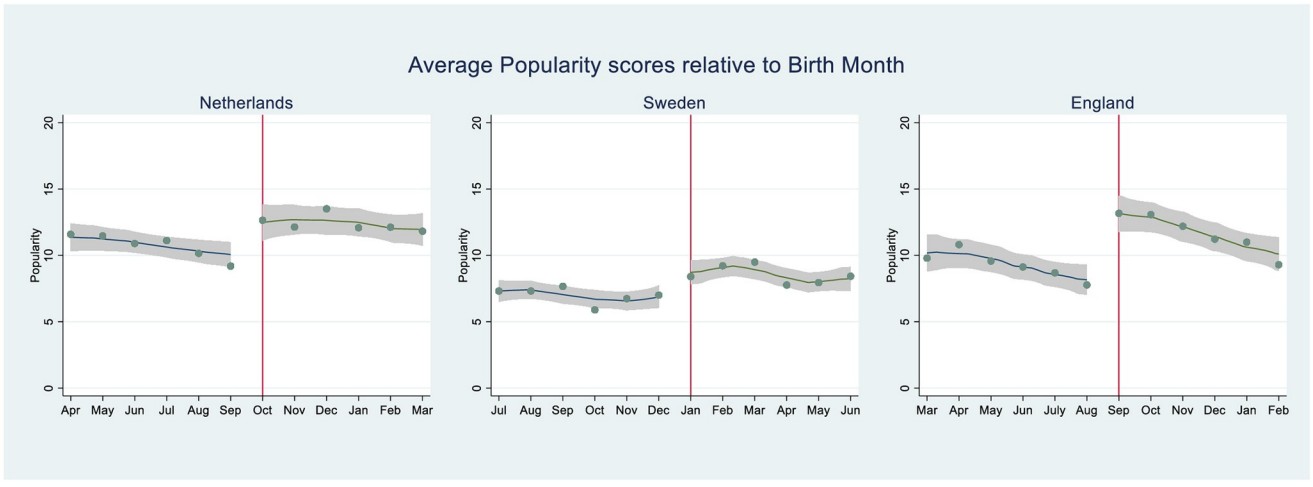

**Fig 1. Average popularity scores of pupils (range 0–100); plotted by distance of birth month to school admission cut-off date.** Grey area visualizes the 95 per cent confidence interval. Source: CILS4EU, wave 1.

**Table 3. Three-level random intercept regression of popularity in class.**

| Variables | Full sample | | Netherlands only | |
|---|---|---|---|---|
| | Model 1 | Model 2a | Model 2b | Model 3 |
| Country (ref = Netherlands) | | | | |
| Sweden | -0.365 | 1.521** | | |
| | (0.611) | (0.680) | | |
| England | -0.971 | 0.899 | | |
| | (0.607) | (0.684) | | |
| Boy | 1.582*** | 1.506*** | 1.869*** | 1.683*** |
| | (0.260) | (0.260) | (0.514) | (0.511) |
| Class size | -0.541*** | -0.542*** | -0.699*** | -0.703*** |
| | (0.024) | (0.024) | (0.051) | (0.052) |
| Class popularity (mean) | 5.046*** | 5.057*** | 5.155*** | 5.140*** |
| | (0.174) | (0.174) | (0.368) | (0.366) |
| % Immigrants in school (ref = 0–10) | | | | |
| 10–30 | 0.133 | 0.140 | 0.590 | 0.546 |
| | (0.393) | (0.392) | (0.764) | (0.760) |
| 30–60 | 0.299 | 0.299 | 0.211 | 0.093 |
| | (0.409) | (0.409) | (0.787) | (0.785) |
| 60–100 | 0.234 | 0.230 | -0.454 | -0.679 |
| | (0.438) | (0.437) | (0.908) | (0.907) |
| Independent school | 1.921** | 1.895** | | |
| | (0.882) | (0.881) | | |
| Past relative age | 0.235*** | 0.097 | 0.096 | 0.165** |
| | (0.065) | (0.069) | (0.078) | (0.084) |
| Past relative age * Sweden | -0.002 | 0.160 | | |
| | (0.092) | (0.131) | | |
| Past relative age * England | 0.221** | 0.366** | | |
| | (0.093) | (0.159) | | |
| Current relative age | | 5.168*** | 5.135*** | 2.198* |
| | | (0.792) | (0.896) | (1.216) |
| Current relative age * Sweden | | -5.508*** | | |
| | | (1.514) | | |
| Current relative age * England | | -5.268*** | | |
| | | (1.853) | | |
| Grade repetition (ref = none) | | | | |
| Primary | | | | -0.082 |
| | | | | (0.819) |
| Secondary | | | | 7.856*** |
| | | | | (1.059) |
| Both | | | | 5.708** |
| | | | | (2.474) |
| Constant | 9.717*** | 7.879*** | 10.901*** | 11.563*** |
| | (0.802) | (0.849) | (1.482) | (1.489) |
| *Number of observations* | | | | |
| Schools (level 3) | 336 | 336 | 100 | 100 |
| Classrooms (level 2) | 681 | 681 | 222 | 222 |
| Individuals (level 1) | 13,251 | 13,251 | 4,308 | 4,301 |
| LL *(df)* | -54600.54 *(16)* | -54579.25 *(19)* | -18264.36 *(12)* | -18201.64 *(15)* |

(*Continued*)

**Table 3.** (Continued)

| Variables | Full sample | | Netherlands only | |
|---|---|---|---|---|
| | Model 1 | Model 2a | Model 2b | Model 3 |
| AIC | 109233.1 | 109196.5 | 36552.73 | 36433.29 |
| BIC | 109352.9 | 109338.8 | 36629.14 | 36528.79 |

Standard errors in parentheses.

\*\*\* p<0.01,

\*\* p<0.05,

\* p<0.1.

category) is 0.235, which means that a pupil born right after the cut-off date in the Netherlands, who was assigned a score of 11 on this variable, had (11 x 0.235) 2.6 percentage points higher score of popularity nominations than a pupil born in the last month prior to the cut-off date (who received score 0). The impact of past relative age is found in each of the three countries.

We also hypothesized about interaction effects between country conditions and the impact of relative age. It was assumed that in England and Sweden, countries with a system of social promotion, past relative age is more strongly associated with current relative age than in the Netherlands, a country with a policy and practice of grade retention. Based on the CILS4EU data, we find a correlation between the two types of relative age of.89 for England,.82 for Sweden, while for the Netherlands it is only.31. These findings validate the assumption that past and current relative age correlate particularly strong in countries with a policy of social promotion.

The interaction terms included in Model 1 reveal that, as hypothesized, the impact of past relative age is stronger in England than in the Netherlands. The birth month effect does not differ between Sweden and the Netherlands. The same pattern is observed in Model 2a. Hence, in England, birth month is stronger related to popularity than it is in the Netherlands, whereas Sweden and the Netherlands do not differ significantly.

Model 2a shows that the impact of current relative age position on popularity is positive and statistically significant in the Netherlands ($b = 5.168$, $p < 0.01$). However, as expected, it appears that the impact of current relative age is significantly less strong in England and Sweden as compared to the Netherlands. In fact, neither in England nor Sweden does current relative age have a significant impact on popularity in models that include past relative age. What matters in these two countries is past relative age, not current relative age.

Further analysis reveals that there is no statistically significant interaction between current and past relative age (S3 Table). We do find that, when analyzing the three countries together, the impact of past relative age is stronger for girls, whereas boys 'gain' more from being relatively older in their current class (S3 Table).

To examine the impact of repetition, we performed additional analyses for the Netherlands only. Model 2b replicates Model 2a, but excludes pupils living in England and Sweden. Model 3 subsequently adds dummy variables for grade repetition. Results from Model 3 indicate that pupils who repeated a grade in secondary education are significantly more popular than those who never repeated a grade. Pupils who repeated a grade in both primary and secondary education are also more popular, whereas those who only repeated a grade in primary education do not differ significantly from those who never repeated a grade. Note that these associations with grade repetition are estimated while keeping constant current relative age position in class.

## Conclusions and discussion

The aim of this study was to investigate whether there exists a Relative Age Effect (RAE) in popularity status of adolescents in class. Two types of relative age were distinguished; next to month of birth with regard to the prevailing cut-off date in primary education, which is the measure for relative age used in previous research [1, 3], the current age composition in class was introduced. Both types of relative age were tested among 14–15 year old pupils from England, the Netherlands and Sweden.

Results confirm that the month of birth in relation to the cut-off date, i.e., *past* relative age, does indeed have a small positive influence on popularity. In other words, youth who were relatively older when starting primary school were more likely to be seen as popular in secondary school. For the concept of *current* relative age, however, we find that this matters in the Netherlands, but not in Sweden and England.

Our study thereby suggests that education policies moderate the relation of relative age with popularity. The contrast is most pronounced between England and the Netherlands. In England, which has a clear system of social promotion in policy and practice, pupils who are the youngest when entering primary school are likely to remain the youngest pupil in their cohort and class. In the Netherlands, grade repetition is common and, so we assume, class skipping, which means that the age composition in class is subject to (yearly) change due to grade repetition and skipping. It also means that in the Netherlands there is a wider age range of students in the classroom, amplifying the (potential) visible age differences between youth and contributing to popularity nominations. This led to the expectation that past relative age would have a stronger association with popularity in England, while current relative age would stronger relate to popularity in Dutch pupils. Results confirm this pattern in the model with both types of relative age.

One policy implication is that teachers in England should become more aware of the enduring impact of the cut-off date when students enter primary school: it affects their well-being [7, 12], popularity and educational outcomes in secondary education [5–9]. In the Netherlands, by contrast, teachers should realize that grade repetition can influence their students beyond their educational outcomes. Those who repeat a grade are more popular than their peers, which can provide them with more self-confidence and status, but also affects the peer hierarchy of the new class that he or she enters when repeating a grade. Based on these cross-sectional findings we cannot make strong arguments for one system over the other, but findings suggest that the impact of relative age depends on educational systems.

Sweden falls in between the extreme cases of England and the Netherlands, the tentative findings from our study suggest. In theory, Sweden has a system of social promotion (like England), but in practice we see that grade retention occurs (like the Netherlands), but not so often. Although we excluded pupils in Sweden who reported to have skipped a class, we suspect that this may not capture all pupils in our dataset who repeated or skipped a grade in Sweden. First, because 'grade repetition' is not in line with the official policy of social promotion. Hence, it may occur more widely than we find (5%); it may be something that happens in practice but without the 'label' of 'grade retention' associated with it. Second, there can also be pupils in Sweden who were kept one year longer in kindergarten, and they may not report that they have repeated a grade. Finally, the data we use did not measure class skipping, so we could not remove those who skipped a class from the data. Thus, although Sweden has a system of social promotion in theory, it appears that in practice schools deviates from these standards, and thereby more-closely resembles the Dutch policy of grade retention and grade skipping.

The current study has several limitations. Due to the cross-sectional design and the accompanying narrow age range, we could not investigate developmental changes in the impact of relative age on popularity. Follow-up studies are encouraged to provide a more in-depth analysis of the mechanisms behind the relations we find. The key theoretical idea of this study is that relative age among children and adolescents make up a relatively large part of their absolute age, and since this is also a period of strong physical and cognitive development [1, 3], the differences in maturity within one classroom can be large. Relative age is perhaps not always known to students in the classroom, but it relates to differences in physiological and cognitive outcomes [1, 3, 22], which subsequently influence the dynamics of competition and status among peers [17, 20]. Future research would benefit from using longitudinal data and investigating the underlying mechanisms that relate relative age to maturity and subsequently to social status and other outcomes of interest.

Another issue is that the study is limited to only three countries and that cross-country differences, as reported here, may or may not be due to educational policies. Follow-up research is encouraged to include more countries to provide stronger tests of such hypothesized policy associations. More generally, we invite scholars to follow up on the institutional approach we take in this study, and to examine how institutional conditions moderate the impact of relative age on outcomes such as popularity, education and sports.

Taken together, this study shows that teachers may benefit from becoming aware of the birth month of their students, because it not only impacts students' educational outcomes, but also their popularity among peers in class -and these two outcomes may be interrelated.

## Supporting information

**S1 Table. Number of cases and age range per birth month after cut-off date.**
(DOCX)

**S2 Table. Fractional probit estimates of popularity in class.**
(DOCX)

**S3 Table. Three-level random intercept models of popularity in class with interactions of past- and current-relative age and gender.**
(DOCX)

## Author Contributions

**Conceptualization:** Danelien A. E. van Aalst, Frank van Tubergen.

**Data curation:** Frank van Tubergen.

**Formal analysis:** Danelien A. E. van Aalst, Frank van Tubergen.

**Methodology:** Danelien A. E. van Aalst, Frank van Tubergen.

**Visualization:** Danelien A. E. van Aalst, Frank van Tubergen.

**Writing – original draft:** Danelien A. E. van Aalst.

**Writing – review & editing:** Danelien A. E. van Aalst, Frank van Tubergen.

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
