## [Decision Letter · Decision Letter 0]

20 Jul 2020

PONE-D-20-13032

More Popular because you’re Older? Relative Age Effect on Popularity among Adolescents in Class

PLOS ONE

Dear Dr. van Aalst,

Thank you for submitting your manuscript to PLOS ONE. After careful consideration, we feel that it has merit but does not fully meet PLOS ONE’s publication criteria as it currently stands. Therefore, we invite you to submit a revised version of the manuscript that addresses the points raised during the review process.

The manuscript has been evaluated by two reviewers, and their comments are available below.

The reviewers have raised a number of concerns that need attention. Reviewer 2 in particular made several comments and requests for an improved framing and more methodological details.

Could you please revise the manuscript to carefully address the concerns raised?

We look forward to receiving your revised manuscript.

Kind regards,

Yann Benetreau

Academic Editor

PLOS ONE

Journal Requirements:

Additional Editor Comments (if provided):

Reviewers' comments:

Reviewer's Responses to Questions

**Comments to the Author**

1. Is the manuscript technically sound, and do the data support the conclusions?

Reviewer #1: Yes

Reviewer #2: Partly

2. Has the statistical analysis been performed appropriately and rigorously? 

Reviewer #1: Yes

Reviewer #2: No

3. Have the authors made all data underlying the findings in their manuscript fully available?

Reviewer #1: Yes

Reviewer #2: Yes

4. Is the manuscript presented in an intelligible fashion and written in standard English?

Reviewer #1: Yes

Reviewer #2: Yes

5. Review Comments to the Author

Reviewer #1: This is a very well written research paper. I really enjoyed reading it. I have just a few minor suggestions for adding some details to make the analysis more meaningful.

1) Can the authors please add a table of age band by month for each country and mention how many cases were in each of these age bands?

2) Can you please add R square value or percentage correctness for each step of the model? Test of significance is not relevant here because the sample is not clearly random selection and you have truncated several class groups below 10 and above 35 cases. P values are increasingly becoming unpopular to show effectiveness of models or hypothesis. Your figure 1 is clearly showing that older students are popular in all contexts (Current Relative Age or Past Relative Age).

3) So what are the policy implications of these findings? What can teachers learn from this and how can education be improved with this knowledge? Please include this in discussion or conclusion of this article.

Reviewer #2: The manuscript “More popular because you’re older? Relative age effects on popularity among adolescents in class” uses a large dataset from multiple countries to examine whether students’ past and current relative age is related to students’ popularity status. The large sample size and novel examination of the association between relative age and popularity are strengths of the paper. There are a few methodological and analytic concerns with the manuscript as well as a need for a stronger theoretical foundation for the paper. Specific recommendations for improvement are outlined below.

1. There is a missed opportunity to thoroughly dive into the popularity literature to discuss what popularity is, what it means, what it gives students, and why the features associated with relative age might translate into popularity (e.g., physical size). For instance, missing from the introduction was a theoretical discussion of how popularity indicates social dominance and prominence among peers, how it confers some social power to those with popularity status, and how the effects of having higher relative age (e.g., physical size, cognitive ability) may yield more social power, dominance, and therefore popularity status. A more robust discussion of these considerations is needed.

2. Is there evidence to suggest popularity is considered the same in those countries? Does it have the same behavioral correlates?

3. The authors introduce the new concept of current relative age and while I do believe it has the potential to contribute to our understanding of students’ adjustment in classrooms, a more robust presentation of this theory is needed. Perhaps some discussion of social comparisons and norm group would be helpful. What does the literature on mixed-aged classrooms (e.g., Montessori structure) say about students’ peer relationships and social roles?

4. The authors indicate that the exact day of birth is known from students in the Netherlands and Sweden but this creates an issue with the relative age variable being created differently for different students depending on their country. I encourage the authors to use the same procedure to calculate relative age using month of birth. If they still want to use the exact day in their calculations, one option would be to run alternative analyses, provided in supplemental materials, which use the more exact day information to see whether and how patterns are the same (i.e., using month of birth versus day of birth). Otherwise, it is problematic to have the same variable calculated with different information.

5. More contextual information is needed about the schools included in the sample. Are these self-contained classrooms where the same group of 14-15 year olds stay with the same group of peers for the entire school day? Or do they switch classrooms (and therefore interact with different peers) for different subjects? Would these students have just transitioned to a new school setting (i.e., secondary/high school)? This information has implications for generalizability and for understanding the proper reference group for the peer nomination measure of popularity (see comment below).

6. Information on the structure of the secondary schools is needed to judge the peer nomination procedure used in this study. Popularity is typically measured with either a limited number of nominations (i.e., 3) in younger school children (i.e., primary schools) because of the self-contained nature of classrooms, hence the voting population is smaller (Cillessen & Marks, 2017). Once students move to larger schools (e.g., secondary schools), they are less likely to be in self-contained classrooms and to reflect this increase in size in the voting population, unlimited nominations are typically allowed for older students. However, there are important cultural differences in these structures. The authors state that students were allowed to nominate up to 5 classmates. Was this constrained to only peers in their self-contained class? Or were they nominating other students in the school?

7. It was good to see the authors accounted for differences in class size by calculating a proportion score for the popularity nomination measure by dividing each students’ total number of nominations by the number of possible nominators (class size). However, the authors should also standardize the nominations by the voting population unit (e.g., classroom or school) per established peer nomination protocol (Estell, Farmer, & Cairns, 2003; Rodkin, Farmer, Pearl, & Van Acker, 2000) to account for the fact that classrooms will have different nomination averages for popularity. As the current study is trying to make general conclusions about the most popular students in each class, in each country, regardless of class size or the popularity average, this standardization piece is important. Thus, both the proportion and standardization procedures are recommended.

8. The authors provide the participation rates for the schools but equally important for the peer nomination measure is the participation rate for classrooms where popularity nominations were gathered. Marks, Babcock, Cillessen, and Crick (2013) recommend that a certain participation rate threshold be met (e.g., at least 40% for very salient nominations, otherwise 60%). Low participation rates threaten the reliability of the nominations. What was the range of classroom participation rates? If there are classrooms with low participation rates, they should be excluded from analyses.

9. Can the author provide the exact language used for the peer nomination measure for popularity in each country? Was the nomination based on free recall or were students provided with class lists? Either way, this information should be included in the manuscript and the implications of either choice should be discussed (i.e., order effect with class lists; Poulin et al., 2011).

10. Information on how data was collected is missing and a brief summary of the data collection procedures is needed. Did students fill out pen and paper surveys or was data collected electronically? When was data collected during the school year? If data was collected right near the start of the school year, this has implications for the development of social reputations (i.e., popularity) among peers, meaning students are still sifting and sorting themselves out and popularity status may still be in flux and this may need to be acknowledged and discussed in the limitations section.

11. The authors state a final sample of 13,014 students but I could not find information about the size of the initial sample and how the initial versus final sample differed. Were their significant demographic differences between those included in the final analyses versus those excluded?

12. I recommend the authors consider including two important interaction effects in their analyses. First, I encourage the authors to examine the interaction between past and current relative age to better understand whether being both historically older (past age) and being oldest in the class currently is associated with higher population (i.e., does being currently older strengthen the association between past relative age and popularity?) or does current relative age not matter as much as being older from the start (past relative age)?

13. Another interaction I was surprised to see not considered was sex differences. Developmentally, 14-15 year olds are likely to be at various stages in pubertal development which will impact characteristics and features often associated with popularity such as physical size, attractiveness, etc. Is the association between relative age and popularity stronger for girl or boys? I strongly recommend the authors consider the role of gender in their analyses.

14. One major concern with the analyses is that the nested nature of the data was not accounted for. The authors should strongly consider multilevel analyses and provide a rationale for their analytic decision.

15. The authors are trying to examine whether different education policies at the country level impact the association between relative age and popularity. They report that 5% of the students in Sweden had repeated a grade, even though they have a policy of social promotion. Were these 5% of students removed from the analyses? If not, this would be problematic because it represents a confound in the analyses between the Netherlands and Sweden. If this subgroup of students were removed, does that change the authors’ expectation that Sweden would fall between England and the Netherlands?

16. Because of the focus on education policy in the paper, I encourage the authors to provide a more robust discussion of how exactly the results have implications for policy. Is one policy “better” than the other? How do they define what “better” means in terms of these associations with popularity? What are the implications for students’ social development depending on each policy?

17. One limitation to acknowledge in the discussion section is that the study cannot speak to the mechanisms driving the association between relative age and popularity. It is unlikely to be age per se, but the associated characteristics, skills, or developmental milestones that come with being older among younger peers. For example, higher past relative age may reflect the compounding impact of higher initial social and language skills which can grow exponentially each year, leading to the type of social savvy associated with popularity. The impact of current relative age may actually reflect differences in physical stature with larger physical bodies being associated with greater dominance and hence, popularity. I would like to see the authors thoughts on these points and some acknowledgement of these possible mechanisms.

References

Cillessen, A. H. N. & Marks, P. E. L. (2017). Methodological choices in peer nomination research. In Peter E. L. Marks & Antonius H. N. Cillessen (Eds.), New Directions in Peer Nomination Methodology. New Directions for Child and Adolescent Development, 157, 21–44

Estell, D. B., Farmer, T. W.,&Cairns, B. D. (2007). Bullies and victims in rural African American youth: Behavioral characteristics and social network placement. Aggressive Behavior, 33, 145–159. doi:10.1002/ab.20176

Rodkin, P. C., Farmer, T. W., Pearl, R., & Van Acker, R. (2000). Heterogeneity of popular boys: Antisocial and prosocial configurations. Developmental Psychology, 36, 14–24. doi: 10.1037/0012-1649. 36.1.14

6. PLOS authors have the option to publish the peer review history of their article (what does this mean?). If published, this will include your full peer review and any attached files.

Reviewer #1: **Yes: **Nadia Siddiqui

Reviewer #2: No

---

## [Author Response · Author response to Decision Letter 0]

12 Jan 2021

We have answered all questions and remarks in the Response to Reviewers letter attached to this submission. Furthermore, the Manuscript with Track Changes shows changes based on the suggestions of reviewers. Since the authors sometimes accepted each others' changes in working on the manuscript, we compared the original and newest version in Word to show the changes that were made to the manuscript.

---

## [Editor Report · Decision Letter 1]

19 Feb 2021

PONE-D-20-13032R1

More Popular because you’re Older? Relative Age Effect on Popularity among Adolescents in Class

PLOS ONE

Dear Dr. van Aalst,

Thank you for submitting your manuscript to PLOS ONE. After careful consideration, we feel that it has merit but does not fully meet PLOS ONE’s publication criteria as it currently stands. Therefore, we invite you to submit a revised version of the manuscript that addresses the points raised during the review process.

I reviewed your original submission (as Reviewer 2) and accepted the offer to serve as Guest Academic Editor for your revision. As Guest Academic Editor, I am tasked with assessing whether the revision meets the publication criteria for PLOS ONE, whether changes to the manuscript adequately addressed reviewer concerns, and whether there are any remaining issues that need to be addressed.

First, the revision meets the following publication criteria: (1) original work (2) that has not been previously published; (3) appropriate statistical analyses performed; (4) the research meets all standards for ethics and research integrity; and (5) the study adheres to appropriate reporting guidelines and standards for availability.

Second, your revision adequately addressed many of the reviewer comments, including my own in my previous role as Reviewer 2. Specifically, the revision addressed Reviewer 1’s suggestion to include information about age band by month per country in a table to aid in readers’ interpretation, provide model fit information, and provide a more robust discussion of the policy implications. In response to my previous comments, I was glad to see how you addressed these major concerns: (1) more robust discussion of popularity and current relative age, (2) more extensive information about the methods and sample, (3) more details about the measures, (4) appropriate statistical analyses to account for the nested nature of the data, and (5) additional exploratory analyses involving interactions between the two age measures and gender. Given the substantial changes made, I can conclude that you were responsive to reviewer feedback.

Third, after reading the revised manuscript, there are a few remaining issues I would like you to address before acceptance related to the following publication criteria: (1) appropriate presentation of the conclusions that are well supported by the data (2) clear writing presented in a logical fashion. 

To improve the flow and organization of this paragraph, I recommend moving the two sentences on p. 3 starting with “Popularity is defined as…” to after the sentence “Popularity and social preference can thus be seen as…” At present, the discussion goes from specifics about popularity, more broadly to types of social status, then back again to specifics about popularity. I would streamline this by starting with the broad discussion of social status and how it’s important to youth then narrowing the focus to the definition of popularity.I appreciated your attempt to respond to my previous comment about behavioral correlates of popularity in these countries by looking at gender difference but gender should not be referred to as a behavioral correlate. Additionally, the intent behind my original comment was to have you review any available evidence of associations between popularity and behavior/ peer experiences (e.g., bullying, victimization) using adolescent samples from those countries to address whether popularity is seen similarly across country contexts. I know of several such studies from the Netherlands (e.g., Cillessen et al., 2014; de Bruyn, Cillessen, & Wissink, 2010) but cannot immediately recall studies using English or Swedish adolescent samples. Perhaps there are some looking at popularity and bullying/ victimization that you can identify. In the absence of any available evidence, please include a discussion in the limitation/ future direction section that more research is needed to verify whether popularity is associated with similar characteristics and behavior in these countries.Clearer statements in your conclusion are needed that describe the results in terms of youth’s popularity status to aid in readers’ interpretation of the findings. Something like: “Youth who were older (i.e., past relative age) when they started school are more likely to be seen as popular when they are in secondary school.”Please temper the causal language in the discussion (e.g., impact).Please include citations to support statements in your conclusion (e.g., that cut-off dates impact students’ well-being and educational outcomes; that physiological differences include dynamics of competition and status among peers, etc).  You mention in the discussion about how class repetition does occur in Sweden sometimes. However, based on the methods (and in response to my previous feedback), you removed these youth who repeated a class from the analyses. Therefore, the results for the Swedish students included in the analyses cannot be explained by the possibility of grade repetition. How else could you interpret this?Please include more discussion about why Sweden and Netherlands did not differ in the association between past relative age and popularity given their different education policies. What could explain their similarity? Are there cultural differences between England versus Sweden and Netherlands that could explain popularity dynamics as it relates to relative age?A more robust discussion is needed for how the association between current relative age and popularity was dependent on the context: Netherland, with its grade repetition policy, may have classrooms with wider range of students, amplifying the (potential) visible differences between youth and contributing to popularity nominations (indeed, a quick visual search of S2 table shows more age ranges spanning 13-17 years).Please include a discussion to help readers interpret the gender differences found and reported in S4 Table: that the positive association between past relative age and popularity was stronger for girls whereas the associations between current relative age and popularity was stronger for boys. What could explain these findings?

Minor changes:

p. 3: change “having influence” to “have influence”

p. 3: Address confusion with the phrase: “in a highly developmental stage.” Do you mean an age when differences in physical development and maturation are more evident? I see that you do later on in the sentence so my recommendation would be to simply remove the phrase “adolescents are in a highly developmental stage”

p. 4: Change the phrasing at the end of the first paragraph to reflect that these statements are expectations. For instance, instead of “they are more likely to be…” perhaps “it’s reasonable to expect them to be…” or “they may be more likely to be perceived as…”.

p. 11: Change “has” to “had” and “has been” to “was” – “An ethical review of…had not been carried out prior to data collection…the CILS4EU project was submitted to the Ethic Committee…”

p. 13: I would not refer to gender as a behavioral correlate of popularity (see comment above).

p. 16: instead of “effect” use “association” to reflect the cross-sectional nature of the study.

p. 28: Table 2. Please remove the Min and Max values for dichotomous variables (country, gender, etc). You may need to reconfigure the table to have a section for continuous variables to report Ms and SDs and another section to report % of the characteristics at each level.

p. 27-34: Please make sure only the final tables remain in the manuscript in the next submission.

p. 21 Table 1: England has the superscript a next to it. Please inform readers what this means in the Note for the table.

References:

Cillessen, A. H., Mayeux, L., Ha, T., de Bruyn, E. H., & LaFontana, K. M. (2014). Aggressive effects of prioritizing popularity in early adolescence. *Aggressive Behavior*, *40*(3), 204-213.

de Bruyn, E. H., Cillessen, A. H., & Wissink, I. B. (2010). Associations of peer acceptance and perceived popularity with bullying and victimization in early adolescence. *The Journal of Early Adolescence*, *30*(4), 543-566.

We look forward to receiving your revised manuscript.

Kind regards,

Molly Dawes, Ph.D.

Academic Editor

PLOS ONE

---

## [Author Response · Author response to Decision Letter 1]

12 Mar 2021

Our detailed response can be found in the response letter to reviewers with a table containing each comment or suggestion and our answer to it.

Dear editors and reviewers, 

Attached please find the revised version of manuscript “More Popular Because You’re Older? Relative Age Effect on Popularity among Adolescents in Class.” We would like to thank you for the detailed reading of our manuscript and the constructive feedback. We made every effort to address each point raised in the revised manuscript and in the present letter. We hope the manuscript now meets the standards for publication in PlosOne.

---

## [Editor Report · Decision Letter 2]

17 Mar 2021

More Popular because you’re Older? Relative Age Effect on Popularity among Adolescents in Class

PONE-D-20-13032R2

Dear Ms. van Aalst,

We’re pleased to inform you that your manuscript has been judged scientifically suitable for publication and will be formally accepted for publication once it meets all outstanding technical requirements.

Kind regards,

Molly Dawes, Ph.D.

Guest Editor

PLOS ONE

Additional Editor Comments (optional):

I appreciate your attention to the few remaining issues and your clear response to my comments.
---

## [Editor Report · Acceptance letter]

12 Apr 2021

PONE-D-20-13032R2 

More Popular Because You’re Older? Relative Age Effect on Popularity among Adolescents in Class 

Dear Dr. van Aalst:

I'm pleased to inform you that your manuscript has been deemed suitable for publication in PLOS ONE. Congratulations! Your manuscript is now with our production department. 

Kind regards, 

on behalf of

Dr. Molly Dawes 

Guest Editor

PLOS ONE